

# Deconstructing the Polymerase Chain Reaction II: an improved workflow and effects on artifact formation and primer degeneracy

Ankur Naqib[1,2], Silvana Poggi[1,3] and Stefan J. Green[1]

[1] Sequencing Core, Research Resources Center, University of Illinois at Chicago, Chicago, IL, United States of America
[2] Rush University, Chicago, IL, United States of America
[3] Northwestern University, Chicago, IL, United States of America

Corresponding author
Stefan J. Green, GreenDNA@uic.edu

## ABSTRACT

Polymerase chain reaction (PCR) amplification of complex microbial genomic DNA templates with degenerate primers can lead to distortion of the underlying community structure due to inefficient primer-template interactions leading to bias. We previously described a method of deconstructed PCR ("PEX PCR") to separate linear copying and exponential amplification stages of PCR to reduce PCR bias. In this manuscript, we describe an improved deconstructed PCR ("DePCR") protocol separating linear and exponential stages of PCR and allowing higher throughput of sample processing. We demonstrate that the new protocol shares the same benefits of the original and show that the protocol dramatically and significantly decreases the formation of chimeric sequences during PCR. By employing PCR with annealing temperature gradients, we further show that there is a strong negative correlation between annealing temperature and the evenness of primer utilization in a complex pool of degenerate primers. Shifting primer utilization patterns mirrored shifts in observed microbial community structure in a complex microbial DNA template. We further employed the DePCR method to amplify the same microbial DNA template independently with each primer variant from a degenerate primer pool. The non-degenerate primers generated a broad range of observed microbial communities, but some were highly similar to communities observed with degenerate primer pools. The same experiment conducted with standard PCR led to consistently divergent observed microbial community structure. The DePCR method is simple to perform, is limited to PCR mixes and cleanup steps, and is recommended for reactions in which degenerate primer pools are used or when mismatches between primers and template are possible.

## INTRODUCTION

The small subunit (SSU) ribosomal RNA (rRNA) gene is the most frequently targeted gene in studies of complex microbial systems. A common approach for microbial community studies is to extract genomic DNA (gDNA) from multiple samples, amplify gDNA by

PCR using locus-specific SSU rRNA gene primers containing sequencing adapters and a sample-specific barcode, and equimolar pooling and sequencing (*Caporaso et al., 2012*). A number of major caveats are associated with such an approach: (i) Microorganisms contain a variable number of rRNA operons (*Klappenbach et al., 2001*; *Angly et al., 2014*) and analyses of rRNA genes present a distorted representation of relative cellular abundance; (ii) PCR primer pools are often degenerate or the primers are anticipated to anneal to template sequences containing mismatches with the primers, thereby producing bias in amplification efficiency among different templates; and (iii) samples are generally heavily amplified (30 cycles or more) leading to a distortion of the template proportions in the original mixtures and to the possibility of extensive chimera formation.

Recently, we identified a novel source of PCR bias—namely, the simultaneous operation of linear copying and exponential amplification during the early cycles of PCR with degenerate primers (*Green, Venkatramanan & Naqib, 2015*). We hypothesized that primer-genomic DNA template annealing operates at a different, and likely lower, efficiency compared to primer-amplicon annealing. These primer-template interactions, operating at different efficiencies, both contribute to distortion of the underlying template community, particularly in the early cycles of PCR. To address this source of bias, we developed the polymerase-exonuclease (PEX) PCR method to separate PCR into two distinct stages of linear copying and exponential amplification. Furthermore, the PEX PCR method prevents the locus-specific primers from interacting with gDNA template after the first two cycles of linear copying. Although effective, the PEX PCR method requires an enzymatic step (exonuclease), which lengthens the workflow. We sought to improve upon the prior protocol and remove the effort associated with exonuclease treatment. Nonetheless, the PEX PCR method—and the separation of linear copying and exponential amplification—serves as the conceptual foundation for the new method. In PEX PCR, after two cycles of linear amplification with locus-specific primers containing 5′ non-degenerate linker sequences, the initial stage of the reaction is terminated, primers are removed with exonuclease I treatment, and the linear copies subsequently amplified using non-degenerate primers targeting the 5′ linker sequences (Fig. 1). Here, we present a method that replaces exonuclease treatment with size-selective bead-based purification (e.g., AMPure XP beads) but achieves substantial savings in overall labor and sample manipulation by a pooling of all samples prior to purification. We note that the strategy of linear copying followed by exponential amplification using primers targeting linker sequences has been employed previously for sequence-independent amplification of whole genome or metagenome DNA using random primers in place of locus-specific primers (*Bohlander et al., 1992*).

The primary objective of this study was to develop an improved pipeline for utilizing the PEX PCR concept, while retaining the ability to reduce PCR bias. To demonstrate the effectiveness of the updated workflow, we replicated a temperature-gradient analysis of a single complex environmental genomic DNA sample using both standard PCR and DePCR workflows. Data were interrogated to examine the observed microbial community structure by method and reaction annealing temperature. In addition, primer utilization profiles (PUPs) were analyzed to assess the effects of annealing temperature on the relative utilization of each primer within a degenerate pool of primers. Subsequently, we examined
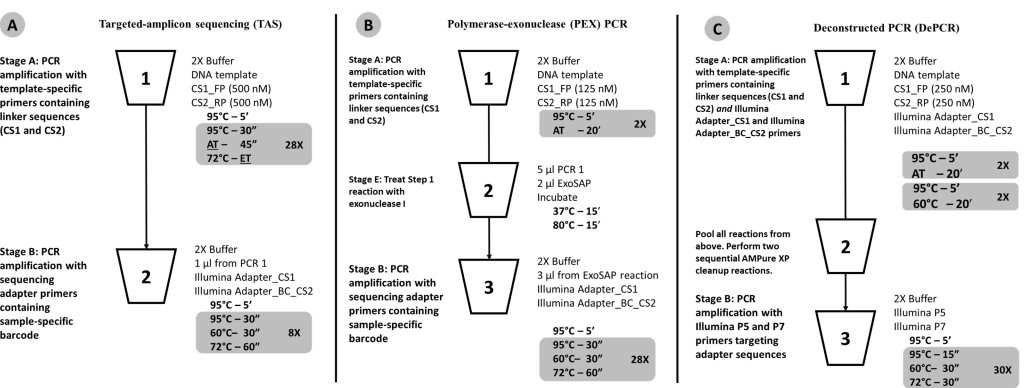

**Figure 1** Schematic of (A) standard (TAS), (B) polymerase-exonuclease (PEX) PCR, and (C) Deconstructed PCR (DePCR) workflows. AT, annealing temperature; ET, Elongation time. CS1, common sequence 1 linker. CS2, common sequence 2 linker. BC, barcode. FP, Forward primer. RP, Reverse primer. Primer sequences are shown in Fig. 2 and Table 1.

the behavior of the amplification system with varying input gDNA levels. A final experiment examined the ability of each unique primer within a degenerate primer pool to amplify a complex environmental sample using both the standard PCR and DePCR methodologies.

# MATERIALS AND METHODS

## DNA templates

A single microbial genomic DNA (gDNA) sample obtained from chinchilla feces was used for this study. Chinchilla feces were acquired from the cage of pet animals and the sample serves as a representative complex microbial community. No specific permissions were required for collection of the chinchilla feces. The fecal sample was extracted using the PowerSoil DNA extraction kit (Mo Bio Laboratories, Carlsbad, CA, USA).

## Primer synthesis

The primers used for this study are 341F (CCTACGGGAGGCAGCAG) (*Muyzer, De Waal & Uitterlinden, 1993*; *Caporaso et al., 2011*) and 806R (GGACTAC**HV**GGGT**W**TCTAAT) (*Caporaso et al., 2011*; *Walters et al., 2016*). The 806R primer pool is 18-fold degenerate, with theoretical melting temperatures ranging from 54.7 °C to 61 °C. Melting temperatures of the primers were calculated using the OligoAnalyzer3.1 tool (*Owczarzy et al., 2008*), assuming 250 nM primer concentration, 2 mM Mg$^{2+}$, and 0.2 mM dNTPs. Synthesis of the primers was performed either as single degenerate primer pools (standard approach), or as individual primers without degeneracies by Integrated DNA Technologies (IDT; Coralville, IA). Primers were synthesized as LabReady and ordered at a fixed concentration of 100 micromolar. Primers contained common sequence linkers (CS1 and CS2) at the 5′ ends, as shown in Table 1. Linker sequences are required for the later incorporation of Illumina sequencing adapters and sample-specific barcodes.

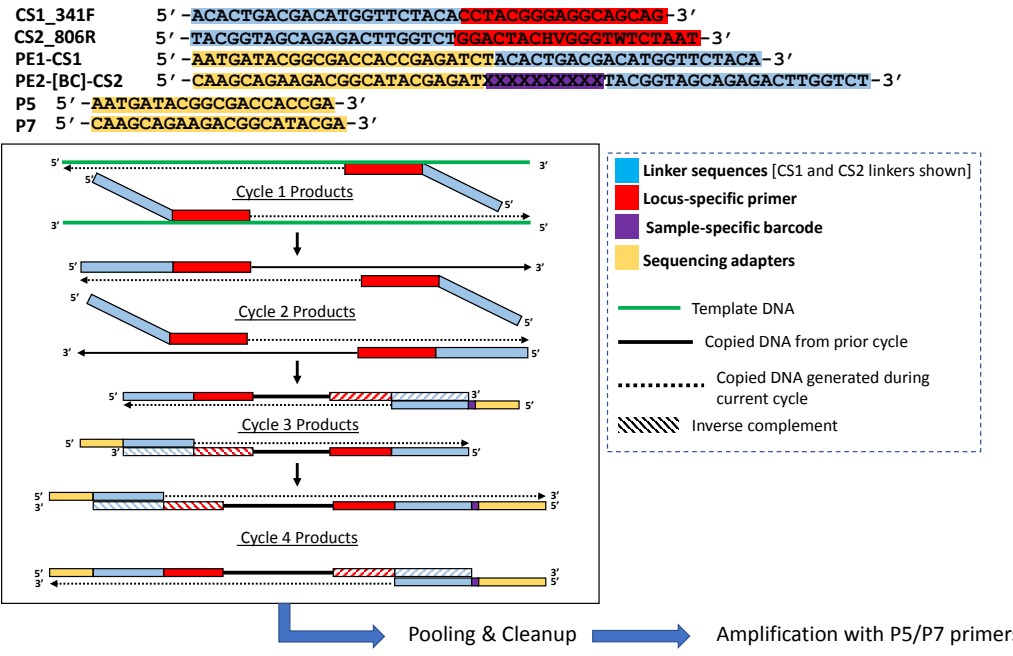

**Figure 2** **Polymerase-generated intermediates in the first stage ("Stage A") of the DePCR workflow.**
Polymerase-extension products generated during the first four cycles of the first stage of the DePCR are
shown. After four linear cycles of copying, the first stage is terminated, samples are pooled and purified,
and subsequently amplified with Illumina adapter primers. Primers used in this study are shown at the top
of the figure, with different functional regions color coded. Red regions represent locus-specific portions
of primers. Blue regions represent linker portions of primers. Yellow regions represent Illumina adapter
sequences. Purple regions represent a variable, sample-specific barcode. Dotted lines represent nucleotide
incorporation by DNA polymerase.

## Standard PCR protocol

The standard PCR protocol or targeted amplicon sequencing (TAS) protocol is a two-stage
NGS library preparation protocol for generating barcoded amplicons ready for Illumina
sequencing, and was performed as described previously (*Naqib et al., 2018*) (Fig. 1A).
Briefly, gDNA was amplified by PCR with primers CS1_341F and CS2_806R. The first
stage PCR reaction was conducted in a total reaction volume of 10 μl. Each reaction
contained 5 μl of MyTaq HS master mix (Bioline, Taunton, MA, USA), 0.5 μl of each
primer or degenerate primer at a concentration of 5 μM (e.g., CS1_341F and CS2_806R;
leading to a 250 nM working concentration), 10 ng of gDNA template, and water up to 10 μl
total volume. The first stage of the PCR was conducted using the following thermocycling
conditions: 95 °C for 5 min, followed by 28 cycles of 95 °C for 30 s, annealing temperature
(from 40 °C to 60 °C) for 30 s, 72 °C for 30 s; and a final elongation step at 72 °C for
7 min. Subsequently, a second PCR amplification was performed in 10 μl reactions in
96-well plates to incorporate Illumina sequencing adapters and a sample-specific barcode.
A mastermix for the entire plate was made using the MyTaq HS 2X mastermix. Each well
received a separate primer pair with a unique 10-base barcode, obtained from the Access
Array Barcode Library for Illumina (Item: 100-4876; Fluidigm, South San Francisco, CA,

**Table 1  Primers used in this study.** Locus-specific primers were synthesized with linker sequences to allow for two-stage PCR amplification and incorporation of sample-specific barcodes, as described in the text. Primer 806R is 18-fold degenerate, and variants were synthesized as a pool as well as individually. Access Array primer sequences, synthesized by Fluidigm (PE1-CS1 and PE2-[BC]-CS2), are shown in Fig. 2.

| 341F primer | Primer sequence | Linker (CS1) sequence | Final sequence name | Final sequence ordered |
|---|---|---|---|---|
| 341F | CCTACGGGAGGCAGCAG | **ACACTGACGACATGGTTCTACA** | >CS1_515F | **ACACTGACGACATGGTTCTACA**CCTACGGGAGGCAGCAG |

| 806R primer and variants | Primer sequence | Linker (CS2) sequence | Final sequence name | Final sequence ordered |
|---|---|---|---|---|
| 806R | GGACTAC**HV**GGG**TW**TCTAAT | **TACGGTAGCAGAGACTTGGTCT** | >CS2_806R | **TACGGTAGCAGAGACTTGGTCT**GGACTACHVGGGTWTCTAAT |
| 806R-RPV1 | GGACTACTAGGGTATCTAAT | **TACGGTAGCAGAGACTTGGTCT** | >CS2_806R_V1 | **TACGGTAGCAGAGACTTGGTCT**GGACTACTAGGGTATCTAAT |
| 806R-RPV2 | GGACTACTAGGGTTTCTAAT | **TACGGTAGCAGAGACTTGGTCT** | >CS2_806R_V2 | **TACGGTAGCAGAGACTTGGTCT**GGACTACTAGGGTTTCTAAT |
| 806R-RPV3 | GGACTACAAGGGTATCTAAT | **TACGGTAGCAGAGACTTGGTCT** | >CS2_806R_V3 | **TACGGTAGCAGAGACTTGGTCT**GGACTACAAGGGTATCTAAT |
| 806R-RPV4 | GGACTACAAGGGTTTCTAAT | **TACGGTAGCAGAGACTTGGTCT** | >CS2_806R_V4 | **TACGGTAGCAGAGACTTGGTCT**GGACTACAAGGGTTTCTAAT |
| 806R-RPV5 | GGACTACCAGGGTATCTAAT | **TACGGTAGCAGAGACTTGGTCT** | >CS2_806R_V5 | **TACGGTAGCAGAGACTTGGTCT**GGACTACCAGGGTATCTAAT |
| 806R-RPV6 | GGACTACAGGGGTATCTAAT | **TACGGTAGCAGAGACTTGGTCT** | >CS2_806R_V6 | **TACGGTAGCAGAGACTTGGTCT**GGACTACAGGGGTATCTAAT |
| 806R-RPV7 | GGACTACTGGGGTATCTAAT | **TACGGTAGCAGAGACTTGGTCT** | >CS2_806R_V7 | **TACGGTAGCAGAGACTTGGTCT**GGACTACTGGGGTATCTAAT |
| 806R-RPV8 | GGACTACTCGGGTATCTAAT | **TACGGTAGCAGAGACTTGGTCT** | >CS2_806R_V8 | **TACGGTAGCAGAGACTTGGTCT**GGACTACTCGGGTATCTAAT |
| 806R-RPV9 | GGACTACACGGGTATCTAAT | **TACGGTAGCAGAGACTTGGTCT** | >CS2_806R_V9 | **TACGGTAGCAGAGACTTGGTCT**GGACTACACGGGTATCTAAT |
| 806R-RPV10 | GGACTACTCGGGTTTCTAAT | **TACGGTAGCAGAGACTTGGTCT** | >CS2_806R_V10 | **TACGGTAGCAGAGACTTGGTCT**GGACTACTCGGGTTTCTAAT |
| 806R-RPV11 | GGACTACCAGGGTTTCTAAT | **TACGGTAGCAGAGACTTGGTCT** | >CS2_806R_V11 | **TACGGTAGCAGAGACTTGGTCT**GGACTACCAGGGTTTCTAAT |
| 806R-RPV12 | GGACTACAGGGGTTTCTAAT | **TACGGTAGCAGAGACTTGGTCT** | >CS2_806R_V12 | **TACGGTAGCAGAGACTTGGTCT**GGACTACAGGGGTTTCTAAT |
| 806R-RPV13 | GGACTACTGGGGTTTCTAAT | **TACGGTAGCAGAGACTTGGTCT** | >CS2_806R_V13 | **TACGGTAGCAGAGACTTGGTCT**GGACTACTGGGGTTTCTAAT |
| 806R-RPV14 | GGACTACACGGGTTTCTAAT | **TACGGTAGCAGAGACTTGGTCT** | >CS2_806R_V14 | **TACGGTAGCAGAGACTTGGTCT**GGACTACACGGGTTTCTAAT |
| 806R-RPV15 | GGACTACCGGGGTATCTAAT | **TACGGTAGCAGAGACTTGGTCT** | >CS2_806R_V15 | **TACGGTAGCAGAGACTTGGTCT**GGACTACCGGGGTATCTAAT |
| 806R-RPV16 | GGACTACCCGGGTATCTAAT | **TACGGTAGCAGAGACTTGGTCT** | >CS2_806R_V16 | **TACGGTAGCAGAGACTTGGTCT**GGACTACCCGGGTATCTAAT |
| 806R-RPV17 | GGACTACCGGGGTTTCTAAT | **TACGGTAGCAGAGACTTGGTCT** | >CS2_806R_V17 | **TACGGTAGCAGAGACTTGGTCT**GGACTACCGGGGTTTCTAAT |
| 806R-RPV18 | GGACTACCCGGGTTTCTAAT | **TACGGTAGCAGAGACTTGGTCT** | >CS2_806R_V18 | **TACGGTAGCAGAGACTTGGTCT**GGACTACCCGGGTTTCTAAT |

| Illumina primers | Final sequence ordered |
|---|---|
| P5 | AATGATACGGCGACCACCGA |
| P7 | CAAGCAGAAGACGGCATACGA |

USA). These Access Array primers contained the CS1 and CS2 linkers at the 3′ ends of the oligonucleotides. One µl of reaction mixture from the first stage amplification was used as input template for the second stage reaction, without cleanup. Cycling conditions were as follows: 95 °C for 5 min, followed by eight cycles of 95 °C for 30″, 60 °C for 30″ and 72 °C for 30″. A final, 7-min elongation step was performed at 72 °C. Samples were pooled and sequenced on an Illumina MiSeq employing V2 chemistry and 2x250 base reads.

## Deconstructed PCR (DePCR) Protocol

As with the TAS method, the DePCR method is also a two-stage PCR process (Fig. 1C) and is a modification of the previously described PEX PCR method (Fig. 1B). For each sample, the first stage reaction of DePCR (four total cycles) was conducted in a 96-well plate with each well containing 5 µl of MyTaq master mix, 0.5 µl of each primer or degenerate primer

at a concentration of 5 μM (e.g., CS1_341F and CS2_806R; leading to a 250 nM working concentration), 10 ng of template, 1 μl Access Array Barcode Library containing a unique sample-specific barcode, and water up to 10 μl. The thermocycler conditions for first stage were composed of two cycles of denaturation at 95 °C for 5 min and annealing (40 °C–60 °C, depending on experiment) for 20 min, followed by two cycles of denaturation for 5 min at 95 °C and annealing at 60 °C for 20 min, and a final extension temperature of 72 °C for 10 min. For temperature gradient experiments, annealing temperatures of 40 °C, 45 °C, 50 °C, 55 °C, and 60 °C were tested. For single reverse primer variant (RPV) analyses, an annealing temperature of 50 °C was used for both TAS and DePCR amplification reactions. Subsequently, a pool composed of 5 μl from the first reaction of each sample was collected and processed for cleanup using AMPure XP beads (Beckman-Coulter) at 0.7X per the manufacturer's recommendations. The cleaning step was performed twice, sequentially. A final elution volume of 20 μl was used to concentrate the sample prior to the second stage of the DePCR reaction. The second stage reactions were conducted in a final volume of 20 μl; the reaction contained 10 μl of MyTaq HS master mix, 1 μl of Illumina P5 (AATGATACGGCGACCACCGA) and P7 (CAAGCAGAAGACGGCATACGA) primers, 2 μl of purified template from pooled first stage PCR, and water up to 20 μl. The thermocycler conditions were: 95 °C for 3 min, 30 cycles at 95 °C for 15 s, 60 °C for 30 s and 72 °C for 30 s. Prior to sequencing the pooled libraries were purified using a Pippin Prep DNA Size Selection System (Sage Science), employing a 2% agarose gel cassette and selecting for fragment sizes from 450-600 bp. Sequencing of the amplified pool was performed on an Illumina MiSeq employing V2 chemistry and 2x250 base reads, and demultiplexing of sequence data was performed on instrument. Library preparation and sequencing were performed at the UIC Sequencing Core (UICSQC).

## Sequence data analysis

Raw sequence FASTQ files were merged using the software package PEAR (*Zhang et al., 2013*), with default parameters. For analysis of primer utilization profiles, merged sequences were trimmed using the software package trimmomatic (*Edgar, 2010*), and sequences shorter than 400 bases and longer than 500 bases were removed. Using Unix bash scripting, exact primer sequences were searched for within these trimmed sequences and counted. For microbial community analysis, PEAR-merged sequences were initially processed through the software package CLC genomics (v10; Qiagen, Aarhus, Denmark) to remove primer sequences, to perform quality trimming (below Q20 removed), and size trimming (below 400 bases removed). Sequences were then screened for chimeras using the USEARCH61 algorithm (*Edgar, 2010*), and putative chimeric sequences were removed from the dataset. Subsequently, sequences were pooled and clustered into operational taxonomic units (OTUs) at a threshold of 97% similarity (QIIME v1.8.0) (*Caporaso et al., 2010*). Representative sequences from all OTUs were annotated using the UCLUST algorithm and the Greengenes 13_8 reference database (*McDonald et al., 2012b*), and a biological observational matrix (BIOM) was generated by this annotation pipeline (*McDonald et al., 2012a*). The BIOM file was analyzed and visualized using the software package Primer7 (*Clarke & Gorley, 2015*) and the R environment (*R Core Team,*

*2013*). The R package 'vegan' (*Oksanen et al., 2011*) was employed to generate alpha diversity indices (Shannon, richness, and evenness indices) and to produce rarefied BIOM files. Bray–Curtis dissimilarity indices were calculated within the R package 'vegan' and these indices were used to evaluate differences in composition between samples. Analysis of similarity (ANOSIM) calculations were performed at the taxonomic level of genus, using square root transformed data. Initial analysis and processing of the samples was performed using QIIME (v1.8.0) package scripts. Metric multi-dimensional scaling (mMDS) plots were generated using the cmdscale and ggplot2 functions (*Wickham, 2016*) within the R programming environment. Ellipses, representing a 95% confidence interval around group centroids, were drawn assuming a multivariate t-distribution. Some visualizations were performed using the software package OriginPro 2018 (OriginLab, Northampton, MA, USA). Rarefaction and group-significance testing (i.e., non-parametric Kruskal-Wallis test) were performed using the QIIME software package.

## Data archive

Raw sequence data files were submitted in the Sequence Read Archive (SRA) of the National Center for Biotechnology Information (NCBI). The BioProject identifier of the samples is PRJNA506229. Full metadata for each sample are provided in Table S1.

# RESULTS

## Theory

The Deconstructed PCR (DePCR) method is based on the polymerase-exonuclease (PEX) PCR method described previously (*Green, Venkatramanan & Naqib, 2015*). We previously noted that the first two cycles of PCR are unique in that no amplification of the template is performed. Rather, linear copying of the template nucleic acid prepares the reaction for exponential amplification, starting in the third cycle. In the prior manuscript, linear copying of the original gDNA template was separated from exponential amplification of target copies using exonuclease I (Fig. 1B). Locus-specific primers containing 5′ linker sequences anneal to genomic DNA during two cycles of amplification. Subsequently, exonuclease I was used to remove unused primers from reaction mixtures. Finally, the copied templates were exponentially amplified using primers targeting the 5′ linker sequences but not the source genomic DNA. This approach is viable, but cumbersome due to the need for exonuclease treatment of each sample, and for individual amplification of each sample with primers containing Illumina sequencing adapters and sample-specific barcodes.

We modified the original PEX PCR protocol by including both locus-specific primers containing 5′ linkers as well as primers with Illumina sequencing adapters, sample-specific barcodes, and 3′ linkers together in the first linear stage of the DePCR reaction (Figs. 1C and 2). Thus, the DePCR approach combines primer sets used in both stage A and B of the PEX PCR method in the same reaction. In addition, four cycles of linear copying are performed, instead of two as in the PEX PCR method (Figs. 1 and 2). The resulting products are target copies containing Illumina sequencing adapter sequences, sample specific barcodes, linker sequences, and the region of interest. The four cycles of copying serve to prepare the
templates for exponential amplification but also (unlike PEX PCR) incorporate a sample-specific barcode so that samples can be pooled and amplified exponentially simultaneously in the second stage. As with PEX PCR, the linear amplification stage—if operating at 100% efficiency—does not increase the total number of targets from that present in the source template DNA.

After linear copying during the first four cycles, the reactions are pooled and purified to remove unincorporated primers. It is essential for the proper functioning of the method that the primers from the initial stage of the reaction are completely removed; otherwise these locus-specific primers continue to interact with template and amplicons during exponential amplification cycles. We observed that a single cleanup using AMPure XP beads (0.7X) was not sufficient to fully remove all primers; therefore, a double cleanup (i.e., two sequential AMPure XP 0.7X cleanups of the pooled reactions) is performed. The final purified DNA includes a range of DNA types, but only the fragments that contain Illumina sequencing adapters at both ends of the molecule have been generated only through linear copying steps and are available for amplification using Illumina P5 and P7 primers (Fig. 2). The entire pool is then used as input template for subsequent amplification using primers consisting of Illumina P5 and P7 sequences. Linear-copied DNA fragments from all samples within the pool, each now containing a sample-specific barcode, are thus subject to exponential amplification simultaneously. One useful feature of this approach is that hundreds of samples can be amplified simultaneously within a single reaction. The theoretical advantages of this novel workflow include: (1) the elimination of a separate exonuclease step for each sample, (2) the rapid reduction of many reactions into a single reaction for purification and exponential amplification, and (3) all associated benefits of the prior PEX PCR, in which linear and exponential amplification stages of PCR are isolated from each other and where locus-specific primers are only active for two linear cycles of copying.

## Validation of the DePCR method

To assess the effects of amplification method (TAS vs DePCR) and annealing temperature on observed microbial community structure, a single genomic DNA sample was amplified across multiple annealing temperatures using both amplification strategies. Five technical replicates for each condition were performed, and amplicons were sequenced together. The data were analyzed to determine if there were significant differences in sequence metrics (chimera formation), alpha diversity (richness and Shannon index), and observed community structure (beta diversity analyses including multi-dimensional scaling, analysis of similarity (ANOSIM), and taxon-level group-significance testing). Rates of detectable chimera formation were several orders of magnitude lower with the DePCR pipeline relative to the TAS pipeline, regardless of annealing temperature (Table 2). Average chimera detection rate for TAS-processed samples range from 5.16 to 6.53%, while that for DePCR-processed samples ranged from 0.03–0.1%; this difference was significant at all annealing temperatures tested (ANOVA, $P < 0.001$). Low rates of detectable chimeras were found in all experiments conducted with DePCR, with averages in the range of 0.01–0.1% (Table 2). Conversely, alpha diversity metrics (genus-level richness and Shannon index),

**Table 2  Rates of detectable chimeras in sequence data.** Average rates of detectable chimeras are shown for each experiment performed in this study. Significantly lower rates of chimera formation were observed for DePCR-amplified gDNA samples relative to TAS-amplified samples, across multiple annealing temperatures. No significant difference in chimera formation was observed with DePCR methodology with varying gDNA input levels. Significantly higher chimera formation was also observed with TAS relative to DePCR when individual primer variants (RPVs) were utilized.

| Experiment | PCR Method | Annealing Temp. (°C) | Input concentration (ng/reaction) | Chimera detection rate [Average (SD)] | ANOVA |
|---|---|---|---|---|---|
| | TAS | 40 | 10 | 5.16% (0.37%) | 1.41E-09 |
| | DePCR | 40 | 10 | 0.05% (0.03%) | |
| | TAS | 45 | 10 | 6.49% (0.29%) | 4.05E-11 |
| | DePCR | 45 | 10 | 0.10% (0.07%) | |
| Annealing temperature | TAS | 50 | 10 | 6.53% (0.21%) | 2.02E-12 |
| | DePCR | 50 | 10 | 0.04% (0.02%) | |
| | TAS | 55 | 10 | 5.69% (0.39%) | 9.66E-10 |
| | DePCR | 55 | 10 | 0.05% (0.02%) | |
| | TAS | 60 | 10 | 5.46% (0.49%) | 7.56E-09 |
| | DePCR | 60 | 10 | 0.03% (0.02%) | |
| | DePCR | 50 | 20 | 0.05% (0.02%) | |
| | DePCR | 50 | 10 | 0.03% (0.03%) | |
| Input gDNA concentration | DePCR | 50 | 5 | 0.03% (0.01%) | 5.20E-01 |
| | DePCR | 50 | 2.5 | 0.02% (0.01%) | |
| | DePCR | 50 | 1.25 | 0.03% (0.03%) | |
| Reverse primer variants | TAS | 50 | 10 | 11.98% (3.85%) | 0.00 |
| | DePCR | 50 | 10 | 0.06% (0.08%) | |

**Notes.**

SD, standard deviation.

were slightly and significantly higher in TAS-based analyses relative to DePCR. Genus-level richness was on average from 1.06–1.21X higher in TAS analyses relative to DePCR, across annealing temperatures from 40 °C to 60 °C (one-way ANOVA; $p$ values ranged from 1.9E-5 to 1.3E-1; Table 3). Shannon indices were from 1.03–1.06X higher in TAS analyses relative to DePCR across annealing temperatures from 40 °C to 60 °C (ANOVA; $p<8.13E-4$; Table 3).

A strong, significant effect of annealing temperature on the observed microbial community structure was seen in both TAS and DePCR amplification methods (Fig. 3A). Although the overall scale of difference between TAS and DePCR was modest (maximum Bray–Curtis dissimilarity between samples = 0.23 between a TAS sample with 60 °C annealing temperature and a DePCR sample with 40 °C), there was a significant effect of amplification method on observed microbial community at all temperatures (Table S2). Two-way ANOSIM analyses indicated significant differences by temperature across methods ($R=0.832$; $p=0.0001$; Fig. 3B), and by amplification method across temperatures ($R=0.988$; $p=0.0001$; Fig. 3C). Similar trends were observed for increases in annealing temperature in both methods, with temperature loading primarily on MDS axis 1. As previously noted (*Green, Venkatramanan & Naqib, 2015*), greater variability in observed microbial community structure was noted with DePCR with low annealing temperature, particularly at 40 °C (Fig. 3A).

**Table 3 Alpha diversity indices of observed microbial communities.** Shannon indices were calculated at the taxonomic levels of genus for all samples amplified using TAS and DePCR methodologies across five annealing temperatures of 40°, 45°, 50°, 55° and 60 °C. Datasets were rarefied to 4,500 sequences/sample. For each methodology and annealing temperature, an average and standard deviation of five technical replicates is shown. At all temperatures, TAS-amplified samples had higher Shannon indices relative to DePCR-amplified samples.

| PCR Method | Annealing Temp. (°C) | Shannon Index [Average (SD)] | ANOVA | Richness [Average (SD)] | ANOVA |
|---|---|---|---|---|---|
| TAS | 40 | 2.69 (0.02) | 4.76E-05 | 61.20 (1.92) | 1.92E-05 |
| DePCR | 40 | 2.55 (0.03) | | 50.60 (1.82) | |
| TAS | 45 | 2.72 (0.03) | 5.86E-05 | 60.60 (2.70) | 1.32E-01 |
| DePCR | 45 | 2.59 (0.03) | | 57.20 (3.63) | |
| TAS | 50 | 2.74 (0.03) | 2.58E-04 | 64.00 (2.65) | 6.56E-02 |
| DePCR | 50 | 2.66 (0.01) | | 59.60 (3.78) | |
| TAS | 55 | 2.72 (0.02) | 8.13E-04 | 62.00 (1.87) | 2.98E-02 |
| DePCR | 55 | 2.64 (0.03) | | 58.60 (2.19) | |
| TAS | 60 | 2.72 (0.01) | 6.16E-04 | 60.60 (2.70) | 3.31E-02 |
| DePCR | 60 | 2.63 (0.03) | | 56.60 (2.19) | |

**Notes.**

SD, standard deviation.

One key feature of the DePCR methodology is the ability to determine which primers in a degenerate pool are interacting with the source genomic DNA. This is achieved as the exponential amplification of the template is performed using primers targeting Illumina sequencing adapters and not the locus-specific primers (Figs. 1 and 2). Locus-specific primers only interact with the gDNA and the first linear copies of gDNA during the first two cycles of the DePCR method. These primer sequences are retained during exponential amplification with primers targeting linker sequences. Conversely, in standard PCR, the locus-specific primers interact with both the genomic DNA template and with copies made from the genomic DNA during exponential amplification; thus, information regarding primer-gDNA template interactions are lost (*Green, Venkatramanan & Naqib, 2015*). We thus examined the so-called "primer utilization profiles" (PUPs) for these reactions (Fig. 4). The relative frequency of each of the 18 unique primer variants is shown for each replicate at each PCR condition (temperature x method). Standard PCR amplification protocol (TAS) removes primer-template interaction information as primer-amplicon interactions throughout the amplification reaction tolerate mismatches; all 18 primer variants are used at similar frequencies, regardless of annealing temperature (Fig. 4A). Some patterning is observed in the TAS method, but overall diversity of primer utilization is extremely high and only small differences were observed between temperatures of 40−60 °C (Fig. 4B). The average Shannon index for PUP profiles of TAS samples across all annealing temperatures was 2.859–2.864; the maximum possible natural log Shannon index for 18 features is 2.890. This PUP diversity profiling demonstrates that for standard TAS PCR, the primers used in copying throughout the amplification reaction are not dependent on annealing temperature.

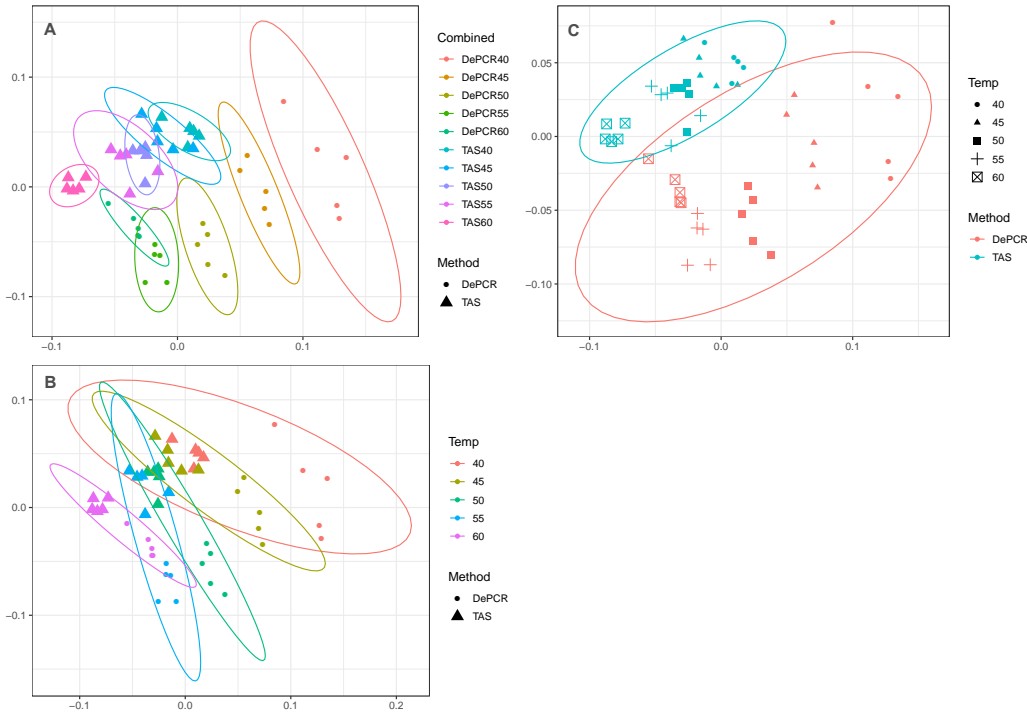

**Figure 3** Effect of PCR methodology and annealing temperature on observed microbial communities. Genus-level abundance data were visualized using metric MDS (mMDS) ordination employing a distance matrix based on Bray–Curtis similarity. For each PCR condition (TAS or DePCR), five technical replicates were analyzed using annealing temperatures of 40°, 45°, 50°, 55° or 60° Celsius. Ellipses represent 95% confidence intervals around centroids. Rarefaction was performed to a depth of 4,500 sequences per sample. Observed community structure was significantly different across (A) all combinations of temperature and method (one-way ANOSIM Global $R = 0.713$; $P = 0.0001$); (B) temperature (two-way ANOSIM $R = 0.832$; $p = 0.0001$), and (C) amplification method (two-way ANOSIM $R = 0.988$; $P = 0.0001$).

Conversely, a strong effect of annealing temperature is observed on the PUP of samples amplified using the DePCR protocol (Figs. 4A and 4B). A shift in PUP patterning is observed with increasing annealing temperature, and at 60 °C two primer variants (RPV5 and RPV15) dominate. At lower annealing temperatures, a broader range of primers are utilized in the initial stages of gDNA copying. The relationship between annealing temperature and primer utilization richness (here represented as the Shannon index) was best fit with a polynomial equation and is shown in Fig. 4C. As annealing temperature increases, fewer and fewer primer variants interact with the gDNA template. Conversely, at the lowest tested annealing temperature of 40 °C, the Shannon index of the DePCR amplicons nearly matched that of the TAS. Several primer variants, however, including RPVs 10, 12, 14 and 18, were poorly utilized in DePCR amplifications regardless of annealing temperature (Fig. 4A). These four variants included variants with high melting temperatures (57.4, 57.5, 58 and 59.8 °C), while the two most utilized RPVs at PCR annealing temperatures of 60 °C had moderate to high annealing temperatures (56.4 and 58.7 °C). Thus, the melting temperature of the primer did not directly correlate with utilization at different PCR annealing temperatures in this system. The observed primer
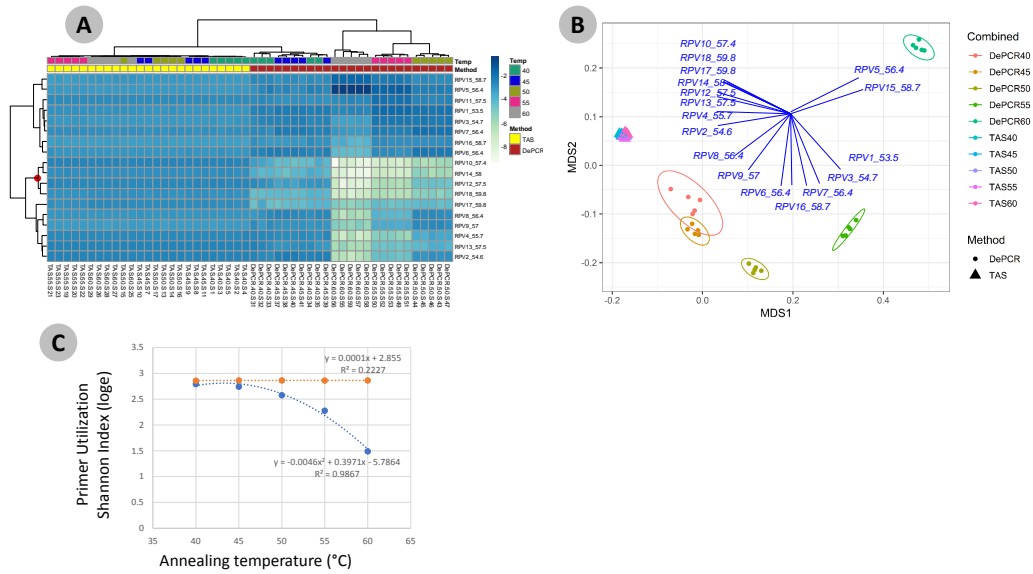

**Figure 4** **Effect of annealing temperature and amplification methodology on primer utilization profiles (PUPs).** (A) Two-way clustered heatmap of log-transformed primer variant utilization during amplification of fecal genomic DNA. Samples (columns) are color-coded by amplification method (TAS or DePCR) and amplification annealing temperature (40°, 45°, 50°, 55° and 60 °C), with five technical replicates per condition and rarefaction to 1,800 sequences/sample. Primers (rows) are clustered by profile similarity across all samples and represent all 18 primer variants (RPV1–RPV18) present in the 806R degenerate primer pool. Theoretical melting temperatures for each primer are shown adjacent to primer name. (B) mMDS ordination of PUPs based on Bray–Curtis similarity. Vectors represent Pearson correlations (>0.9) for each primer variant. Ellipses represent 95% confidence intervals around centroids for DePCR amplification reactions. Five technical replicates per condition were generated and for each sample, rarefaction was performed to 1,800 sequences. (C) Regression analysis was performed was performed on average Shannon index values for primer utilization for each methodology (TAS and DePCR) across annealing temperature. A very small effect of annealing temperature on primer utilization evenness was observed in TAS (orange line). A negative quadratic relationship was observed between annealing temperature and primer utilization evenness in DePCR (blue line). Analyses were based on five technical replicates rarefied to 4,500 sequences per sample.

utilization profiles represent a template-specific phenomenon, and different PUPs would be recovered with different DNA templates.

## Determination of linearity in DePCR amplification

In the DePCR protocol, after four initial cycles of linear copying during the first stage of DePCR, samples are pooled prior to purification and second stage amplification with Illumina P5 and P7 primers. The pooling of samples can only be performed because of the incorporation of a sample-specific unique barcode for each sample during the first stage. During the second stage amplification, primers targeting the Illumina adapters are used for amplification, and all templates from all samples are amplified simultaneously (Fig. 1C). Since there is no opportunity for primer-template bias during the second stage (i.e., **Stage B of** Fig. 1C) of amplification (all amplifiable template molecules contain Illumina sequencing adapters) and primers are non-degenerate, the relative abundance of template molecules from a single sample within the pool should be maintained during

amplification. To determine if the relative abundance of template molecules from each sample was maintained in the DePCR protocol, we performed an experiment in which input gDNA (feces) was varied from 1.25 ng to 20 ng per 10 µl reaction. All input levels were performed with five technical replicates. After the first stage (4 cycles) of the DePCR, all replicates from all gDNA input levels were pooled in equal volume and purified. The purified product was then amplified with P5 and P7 primers, and the final pool sequenced. We first assessed whether the input DNA concentration was correlated with the total number of reads generated using this approach (Fig. S1). Since all samples were amplified together, and low input DNA samples should theoretically provide fewer molecules to the combined pool, we hypothesized that a linear relationship should exist between input DNA in the first stage and the number of reads generated per sample. A significant positive correlation between input gDNA concentration and absolute number of reads recovered from each sample was observed, though substantial variability at each input concentration was observed ($R^2 = 0.58$, Fig. S1C). We also sought to determine if the input gDNA concentration from the same sample had a significant effect on the observed microbial community structure. Although there was a positive correlation between input gDNA and total number of sequences recovered, we observed no significant effect of input gDNA on the microbial community structure (Fig. S1A; Global ANOSIM $R = -0.034$; $p = 0.79$). Similarly, no significant difference in primer utilization was observed with different gDNA input concentrations (Fig. S1B). Thus, increasing input gDNA concentration alters the number of molecules passing to the second stage of the DePCR reaction, but within the measured concentration range did not affect the primer utilization profile or final observed microbial community structure.

## Assessing the effect of individual primers in a degenerate primer pool

Degenerate primer pools are generally used to amplify genomic DNA, although not all primers actively interact with the source gDNA (Fig. 4A). This degenerate mixture of primers is employed to target a broad range of taxa, and the presence of additional primer variants in pools has been shown to improve detection of known microbial lineages (*Hayashi, Sakamoto & Benno, 2004*; *Frank et al., 2008*; *Parada, Needham & Fuhrman, 2016*; *Apprill et al., 2015*). In standard PCR, all primers do eventually interact with amplified copies of gDNA during the many cycles of exponential amplification; however, many primers do not interact with the source genomic DNA due to preferential annealing of other primers (Fig. 4A). We sought, therefore, to determine how much microbial diversity could be detected using each primer variant independently in PCR reactions using both the TAS and DePCR methods. In addition, we sought to determine how the observed microbial community structure differed by single primer variant usage. We hypothesized that the single primer variant PCR would better approximate degenerate primer pools when using the DePCR method relative to the TAS method, as our prior work showed that a deconstructed PCR approach was more tolerant of mismatches between primer and gDNA template than TAS (*Green, Venkatramanan & Naqib, 2015*). The tolerance of mismatches may lead to better capture of microbial community diversity when a greater number of mismatches between primer and template are present, as is expected in a single

primer PCR. To explore this, we PCR-amplified a single gDNA template (feces) with the 18 unique reverse primer variants (RPVs) from the degenerate primer pool. Each reaction was performed in technical duplicates, and each reaction was performed using the DePCR and the TAS method. Three RPVs from the TAS method were removed from the analysis due to pipetting error, as determined by primer utilization profiles. These included one replicate of RPV5 and both replicates of RPV15 (Table S1). We compared alpha and beta diversity analyses of the PCRs employing 15–18 unique RPVs to those generated with the fully degenerate primer set. All alpha and beta diversity analyses were performed on data rarefied to a depth of 1,800 sequences/sample (Table S1—experiment 3).

When employing fully degenerate primer pools, observed alpha diversity (Shannon index) of the fecal sample was slightly, but significantly higher when analyzed using the TAS protocol relative to the DePCR protocol (average Shannon index, five replicates, 2.71 to 2.66; ANOVA $P < 0.001$; Table 4). We then calculated average Shannon indices for analyses of the same gDNA sample with individual RPVs, employing TAS and DePCR protocols. The average Shannon index for the TAS reactions with unique RPVs (2.40) was significantly lower than that measured for the DePCR reactions (2.58) (ANOVA $P < 0.001$; Table 4). Finally, all RPV data, rarefied to 1,800 sequences per sample, was pooled together for TAS and DePCR approaches, independently. These combined datasets were then randomly sub-sampled to 1,800 sequences. These rarefactions were performed five times, and the average Shannon index for the combined RPVs was calculated. In this approach, average Shannon index from TAS (2.48) was significantly lower than for DePCR (2.69) (ANOVA $P < 0.001$; Table 4). Across all three methods of calculating observed diversity, there was no significant difference in measured Shannon index for the DePCR method (ANOVA, $P = 0.377$), while a significant decrease with each RPV independently was observed with the TAS method (ANOVA, $P = 3.69e−8$). When each RPV is used independently in the TAS protocol, the overall captured diversity is lower than with reactions with degenerate pools (Table 4) due to the greater number of potential mismatch interactions that can occur when a complex template is amplified with a single, non-degenerate primer. As the DePCR method is more tolerant of mismatches, no significant decrease in average Shannon index was observed. However, the observed variance in Shannon index among the individual RPVs was greater for the DePCR than for the TAS method (Table 4).

We next examined the structure of the observed fecal microbial communities in standard TAS and DePCR with degenerate primer pools, and with reactions conducted using RPVs (Fig. 5). We observed high reproducibility for five replicates using TAS (i.e., 'TAS_pool') or DePCR (i.e., 'DePCR_pool') with degenerate primer pools (Figs. 5A and 5B) and observed microbial community structure was significantly different between TAS and DePCR employing the degenerate primer pools (ANOSIM, $R = 0.401$, $p = 0.001$). Compared to amplifications with degenerate pools of primers, within-group variability was much greater for the analyses of RPVs individually with either amplification protocol (Figs. 5A and 5B, 'TAS' and 'DePCR'). Within-group Bray–Curtis dissimilarity (BCD) of amplicons from the 15 (TAS) to 18 (DePCR) RPVs ranged from 0.03 to 0.36 for the TAS method and from 0.04 to 0.68 for the DePCR method (ANOVA $P < 0.001$; Fig. 5B). Conversely, the within-group BCD for five technical replicates generated with degenerate primer pools

**Table 4 Effects of amplification method and reverse primer variants on observed microbial community alpha diversity.** Fecal gDNA was PCR amplified with 18-fold degenerate reverse primer pools (5 technical replicates), and with each unique reverse primer variant (RPV; 2 technical replicates). Data sets were rarefied to 1,800 sequences per sample, and Shannon indices (loge) were calculated. When using fully degenerate primer pools, average Shannon index was significantly higher for TAS methodology relative to DePCR methodology. When data from all reactions with individual RPVs were analyzed, average Shannon index was significantly lower for TAS methodology relative to DePCR methodology. Data from RPVs (1,800 sequences/sample) were pooled and re-rarefied to 1,800 sequences (five repetitions), and the resulting average Shannon index was significantly lower for the TAS methodology relative to DePCR methodology. Different approaches with the DePCR method did not generate significantly different Shannon indices (ANOVA $P = 0.377$), while the same approaches generated significantly different Shannon indices (ANOVA $P < 0.001$).

| Comparison | # replicates analyzed | Average Shannon Index (SD), TAS | Average Shannon Index (SD), DePCR | ANOVA |
|---|---|---|---|---|
| Amplification with 18-fold degenerate primer pools | 5 | 2.71 (0.03) | 2.66 (0.04) | 3.14E-05 |
| Amplification with each RPV independently | 33 (TAS) or 36 (DePCR) | 2.4 (0.01) | 2.58 (0.21) | 5.95E-05 |
| Summation of independent RPVs and re-rarefaction to 1800 sequences (5x) | 5 | 2.48 (0.03) | 2.69 (0.02) | 7.40E-07 |
| | **ANOVA** | 3.69E-08 | 3.77E-01 | |

were 0.04 to 0.07 for TAS and 0.05 to 0.11 for DePCR (ANOVA $P < 0.001$). Profiles of the individual RPVs from DePCR analyses could be divided into two groups: (a) RPVs with profiles highly similar to degenerate primer pool analysis with either DePCR or TAS; and (b) RPVs with profiles divergent from the degenerate pool communities, and more similar to RPVs from TAS amplification reactions. Overall, the observed microbial community structure generated using the DePCR method with RPVs and with degenerate pools was not significantly different (ANOSIM $R = -0.306$, $p = 0.99$). Conversely, the observed microbial community structure generated using RPVs was significantly different that that observed with degenerate primer pools for the TAS method (ANOSIM $R = 0.487$; $p = 0.003$). Average BCD between TAS_pool and TAS RPV profiles (0.211) was significantly greater than for DePCR_pool and DePCR RPV (0.154) (ANOVA $P < 0.001$; Fig. 5C). DePCR BCD profiles were heavily weighted toward low dissimilarity, with a long tail of high dissimilarity comparisons. The long tail is a result of some primers generating highly divergent observed microbial communities with the DePCR protocol. Many of the primers which showed the poorest utilization within the degenerate pool (e.g., RPV10, 12, 14, and 18; **node with red dot in** Fig. 4A), generated the most divergent single RPV profiles. This suggests that these primers do not closely match the most dominant taxa within this particular gDNA sample.

# DISCUSSION

We demonstrate here an updated protocol for the Deconstructed PCR methodology (*Green, Venkatramanan & Naqib, 2015*) which reduces the overall complexity of the workflow and increases the throughput. Complete removal of 1st stage (or "Stage A") primers (locus-specific primers containing 5' overhanging linkers) is essential for the effectiveness of the DePCR protocol, and we have replaced the exonuclease step with a bead-based magnetic cleanup. The new method improves throughput by generating barcoded DNA fragments through four cycles of linear amplification; thus, all samples can be pooled before bead-cleanup. This reduces workflow complexity and cost, while retaining the essential features of

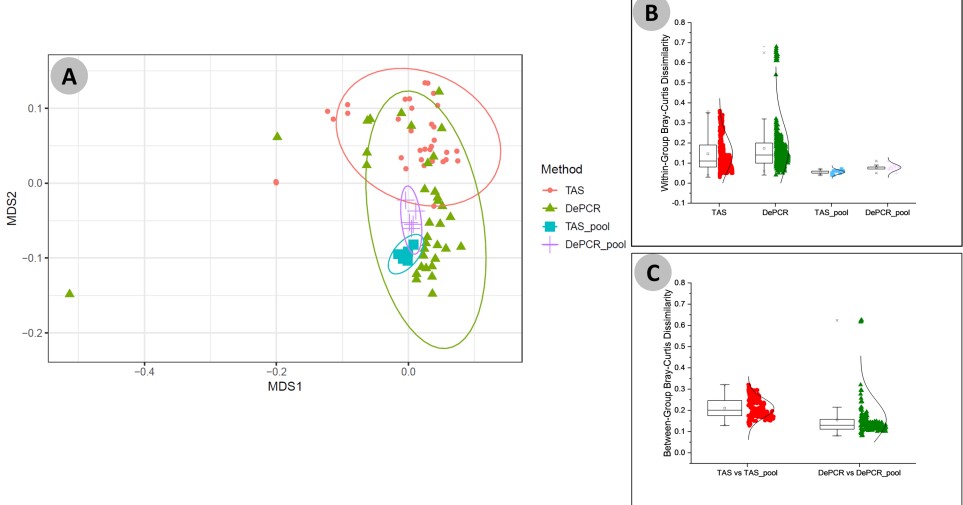

**Figure 5 Microbial community structure revealed using individual primer variants with TAS and De-PCR amplification methodologies.** (A) Fecal gDNA was amplified using the 341F primer with18 unique 806R reverse primer variants (RPVs) under standard PCR (TAS) and DePCR workflows. Three RPVs were removed from the TAS analysis due to pipetting error, as described in the text. Genus-level biological observation matrices (BIOMs) were visualized using mMDS. Each amplification with a unique RPV was performed in technical duplicate, and five technical replicates were generated using degenerate primer pools (TAS_pool or DePCR_pool). All samples were rarefied to 1,800 sequences. Ellipses represent 95% confidence intervals around centroids. TAS profiles generated with RPVs were significantly distinct from TAS profiles generated with degenerate primer pools (ANOSIM $R = 0.487$; $P = 0.003$). DePCR profiles generated with RPVs were not significantly distinct from DePCR profiles generated with degenerate primer pools (ANOSIM $R = -0.306$; $P = 0.99$). (B) Within-group Bray–Curtis dissimilarity distributions for profiles generated with RPVs and with degenerate pools. (C) Between-group Bray–Curtis dissimilarity distributions for observed microbial community structure generated with RPVs and with degenerate primer pools. Average dissimilarity among TAS_pool and TAS RPV profiles (0.211) was greater than for De-PCR_pool and DePCR RPV profiles (0.154) (ANOVA $P < 0.001$).

the DePCR reaction. Complete removal of primers is difficult to measure directly, however; thus, the primer utilization profiles (PUPs) are the clearest indication of successful removal of locus-specific degenerate primers from the first stage of the reaction. With standard PCR, no true signal is obtained from the PUPs, as primer-amplicon interactions during late cycles generates a 'scrambled' signal due to mismatch interactions with amplicons present at high abundance. In DePCR, a PUP signal can be obtained as locus-specific primers only interact with the gDNA template and linear copies during the first two cycles of PCR. Subsequently, all exponential amplification is performed using conserved sequences that are not present in the source gDNA. In this way, the primer sequences interacting with the source gDNA are 'fossilized' and can be interrogated directly. When using this approach, we observed strong effects of annealing temperature on primer-gDNA template interactions, with a negative quadratic correlation between annealing temperature and evenness of primer utilization. At highest annealing temperatures, very few primers from the primer pool anneal to the gDNA template, and this leads to a shift in the sequences that are amplified by PCR with a result of significantly different observed microbial

communities. We note that the elevated annealing temperature by itself does not select for primer variants with the highest theoretical melting temperature. Rather, primer variants, presumably template-specific, are favored regardless of their melting temperature.

A surprising benefit to the DePCR methodology is the reduced rate of chimera formation. Chimeras are artifactual hybrid sequences generated from two or more templates due to incomplete polymerase extension during PCR, and their presence can be difficult to detect and lead to overestimation of diversity and alteration of observed microbial community structure (*Hugenholtz & Huber, 2003*; *Schloss, Gevers & Westcott, 2011*; *Edgar et al., 2011*). Input genomic DNA concentration and target microbial community complexity have been identified as contributors (*Fonseca et al., 2012*; *Lahr & Katz, 2009*). We previously observed that chimera formation was correlated with total number of PCR cycles in both first and second stages of PCR (*Ionescu et al., 2016*), and this has been reported elsewhere in many studies (*Edgar et al., 2011*; *Lahr & Katz, 2009*; *Wang & Wang, 1996*). As many factors can contribute to chimera formation, various solutions have been proposed, including reducing input gDNA concentration (*D'Amore et al., 2016*), reducing PCR cycles (*Suzuki & Giovannoni, 1996*; *Kanagawa, 2003*), employing highly processive enzymes (*Lahr & Katz, 2009*), among others. In this study, we have observed that the use of the DePCR methodology can dramatically and significantly lower rates of observed chimeras resulting in rates that were generally below 0.1%. These low rates of chimera formation were observed across all annealing temperatures and input template concentrations tested. The reasons for the dramatic decrease in chimera formation rate with the DePCR method are likely a result of: (a) reduction in input DNA concentration for exponential amplification due to the double-purification step, (b) higher annealing temperature for the exponential amplification due to targeting of P5/P7 Illumina adapters—potentially reducing the re-annealing of PCR products to other products, and (c) long elongation times during the first cycles, reducing the formation of incomplete molecules during the first stages of PCR. Conceivably, chimera formation with DePCR could be reduced further; we performed 30 cycles of amplification to generate robust PCR yields for sequencing. However, the amplification of the pool of amplicons during the second stage PCR could be titrated across different numbers of cycles, and the reaction with the fewest numbers of cycles yielding sufficient DNA for sequencing could be employed. It is critical to remember that the rate of chimera formation represents only the rate of *detectable* chimera formation, and that chimeras generated from closely related sequences are not only likely to occur at higher rates (*Wang & Wang, 1996*) but are also essentially undetectable by chimera detection software. We note that in this study, amplification of fecal gDNA with degenerate primer pools resulted in higher observed diversity with the TAS method relative to the same sample amplified with the DePCR protocol (Table 4), and this could represent the residual presence of chimeras that were not removed.

*Suzuki & Giovannoni (1996)* previously modeled PCR reactions with mixed templates by incorporating efficiency parameters into equations estimating molarity of amplicon yield. They further estimated second-order kinetics wherein changes in the concentration of specific PCR products alter efficiencies during the amplification, including through inhibition of amplification by competition between primers and amplicons for annealing

locations. With increasing cycle number, reaction efficiency dropped dramatically. The DePCR method theoretically circumvents at least some of these issues. First, since locus-specific primers interact with template only during two cycles of copying (linear only), any differences in amplification efficiency of templates are limited to those two cycles. Subsequently, all templates are amplified with primers targeting sequences common to all amplifiable templates. *Suzuki & Giovannoni (1996)* showed that even a relatively high amplification efficiency could lead to dramatic distortion of the underlying template ratios within 10–15 cycles. In DePCR approaches, amplification efficiency is expected to be lowest during the first two cycles—when primers anneal to gDNA templates with varying numbers of mismatches—and then higher during the remaining cycles as amplification is performed with perfectly matching primers. We also note that in PCRs with degenerate primers, each primer variant is present at a low concentration (total primer concentration/number of variants); in the 2nd stage of the DePCR protocol, a non-degenerate primer at a high concentration relative to each variant is used for amplification. Thus, DePCR limits the number of cycles operating at low primer efficiency and uses high-efficiency reactions to perform exponential amplification. Degenerate locus-specific primer interactions with PCR amplicons are also removed, thereby removing additional variable efficiency annealing steps from the PCR.

We previously demonstrated that a deconstructed PCR approach could help overcome PCR distortions due to mismatches between primers and templates in a mock community (*Green, Venkatramanan & Naqib, 2015*), and we believe this is in part due to the circumventing of multiple cycles with low amplification efficiency. Single mismatches between templates and primers can substantially alter observed microbial community structure, and indeed, many modifications to degenerate primer pools are performed to increase degeneracy by adding single variants targeting specific microbial taxa (*Schloss, Gevers & Westcott, 2011*). In this study, we independently used each primer variant in a degenerate primer pool both to examine the potential for each primer to amplify a complex microbial gDNA template and to assess the ability of the DePCR protocol to enable single non-degenerate primers to broadly amplify microbial taxa with mismatches. We observed that while the observed microbial community structure varied widely using non-degenerate primer variants (both TAS and DePCR), many single non-degenerate primer variants were able to generate reasonable approximations of the microbial community structure as revealed through amplification reactions with degenerate primer pools, thus indicating that the DePCR approach can be used with complex microbial samples to improve tolerance of mismatches. This suggests that a more empirical approach to primer design can be taken by using the DePCR method to reduce the complexity of degenerate primer pools or enable broader target range of highly degenerate primer pools targeting functional genes. Primer utilization profiling can in turn be used to provide empirical evidence demonstrating which primers within the degenerate primer pool are interacting with unknown templates. The inclusion of non-essential variants decreases the concentration of all other primers in a primer pool, and removal of unneeded primer variants may be beneficial. However, when using the same primer set for a broad range of complex genomic DNA samples from

different environments, we expect that the 'essential' primers will vary from system to system.

We can recommend the DePCR protocol for reactions where degenerate primer pools are used or for primer-template systems where mismatches are possible or expected. Several caveats, however, should be considered. First, the method is not recommended for reactions requiring stringent PCR conditions. Second, since reactions are pooled together after the first linear cycles and then amplified, the reactions are sensitive to the relative number of copies within each sample. As observed in Fig. S1C, there is a linear response between input gDNA and number of sequences generated. Thus, input gDNA concentration of similar samples should be carefully controlled to avoid large variance in number of sequences generated per sample. Furthermore, different sample types should be amplified independently, as different sample types may have a different density of targets per ng of DNA, leading to further variance in sequence reads generated. Samples with low input DNA concentrations may require additional cycles of exponential amplification during the second stage of DePCR to generate sufficient product for sequencing. Such samples could be processed independently and then pooled with other sample types prior to sequencing. Third, in the updated DePCR protocol where Illumina P5 and P7 primer are used, polymerase extension copies through the DNA region containing the sample-specific barcode and can introduce errors. In this study, we employed Fluidigm Access Array primers which contain 10-base barcodes with a Hamming distance of 3 (each barcode has at least three mismatches with all other barcodes), and this large Hamming distance should limit mis-assignment of reads. However, with other barcoding systems, or with very high PCR cycle or error-prone polymerases, this source of error could lead to cross-signaling between samples or loss of reads. Finally, we note that when assessing if a DePCR protocol is functioning properly, it is important to employ an analysis of primer utilization across a temperature gradient analysis with standard (TAS) and DePCR workflows. In standard PCR, a small or no effect of temperature should be observed on the PUPs, while a strong shift in primer utilization should be observed with the DePCR protocol. Since primer utilization with DePCR can be extremely broad at low annealing temperatures, it can be difficult to differentiate between a properly operating or failed DePCR protocol without the temperature gradient analysis.

## CONCLUSIONS

We demonstrate here an improved method to reduce bias associated with PCR amplification of complex genomic DNA templates with degenerate primers. The method, DePCR, is a simple and versatile, two-stage PCR protocol allowing highly multiplexed library preparation for Illumina sequencers. When employing this method using a common degenerate primer set targeting microbial 16S rRNA genes, we observed a significant decrease in chimera formation relative to standard PCR amplification. When using non-degenerate primers, the DePCR methodology frequently reduced PCR distortions due to mismatches between primers and templates.

### Funding
The authors received no funding for this work.

### Competing Interests
The authors declare there are no competing interests.

### Author Contributions
- Ankur Naqib conceived and designed the experiments, analyzed the data, prepared figures and/or tables, authored or reviewed drafts of the paper, approved the final draft.
- Silvana Poggi performed the experiments, approved the final draft.
- Stefan J. Green conceived and designed the experiments, analyzed the data, contributed reagents/materials/analysis tools, prepared figures and/or tables, authored or reviewed drafts of the paper, approved the final draft.

### Data Availability
Raw sequence data files are available in the Sequence Read Archive (SRA) of the National Center for Biotechnology Information (NCBI), BioProject PRJNA506229.

### Supplemental Information
Supplemental information for this article can be found online at http://dx.doi.org/10.7717/peerj.7121#supplemental-information.

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
