# Peer review of "Deconstructing the Polymerase Chain Reaction II: an improved workflow and effects on artifact formation and primer degeneracy"

_PeerJ, doi:10.7717/peerj.7121_

## Round 0.1 · original submission · Minor Revisions

Please answer the requests of the reviewers (which are not very extensive and resubmit a revised version).

·

Basic reporting

The paper is clearly and professionally written, with only a few errors. (Please complete the sentence on line 56, and check the grammar on line 297.) References are cited as appropriate, spanning over two decades' worth of method development and refinement. The authors show a comprehensive knowledge of various sources of PCR bias.

Experimental design

The paper provides important but incremental improvements in previously published methods for two stage PCR (TAS and PEX PCR), which were also developed in the same laboratory.

I'd like to make a note that two stage PCR has been around since Bohlander first published the concept in 1992, which was later modified by Joseph Derisi's lab. The original papers had fully degenerate primers for whole genome amplification, instead of partially degenerate primers for targeted amplification. The important point is that 2-stage PCR protocols often have a limited numbers of tagging cycles (2-4) in order to prevent the artifacts that were identified as a "novel source of bias" in the PEX PCR paper.

Validity of the findings

The current method has shown improvements in reducing chimera formation, and the addition of barcodes allows pooling of multiple samples and evaluating the primer utilization profiles of different members of the degenerate primer pool. The addition of a bead-based cleanup step also improves the workflow compared to the restriction enzyme step.

Additional comments

This paper clearly demonstrates how subtle differences in the method can lead to profound differences in the results, which is one of the major criticisms in commercial application of microbial community analysis (e.g. gut microbiome analysis in human health.) In a microbial community, there is an underlying “Truth” to the diversity and abundance of different organisms, and it’s not clear that this method is any closer to revealing the Truth.

The problem with modern microbial ecology is that the 16S databases are full of vestiges of past priming events-- artifacts of priming, cloning, and sequencing over the past 30 years. Have we now “converged” on a universal primer set that is now only 18-fold degenerate? For example, it appears that primer 341F has dropped the degeneracy that was used in earlier studies. (https://doi.org/10.1016/j.mib.2004.04.015)

The bigger picture is that the 16S gene has a number of faults that make it a lousy biomarker, but it's been swept up by a tidal wave of new users and new contributions to public sequence databases. This was a problem that I thought about deeply in my graduate career, because the limitations in the methods obscured the ability to comprehend the underlying biology. (https://doi.org/10.1016/j.mcp.2006.09.003) Now is a different era, though, where high throughput sequencing is inexpensive and accessible. I would love for the authors to consider a new line of inquiry for future method development, which overcomes the biases of the 16S gene.

·

Basic reporting

The article on a novel deconstructed PCR approach, in which the linear and exponential phase of the PCR reaction are separated, is well structured and well written.
The language and phrasing are clear and correct, the referencing and the contextualization of the research are sufficient and appropriate.
Figures and Tables are clear and well structured, and the conclusions are overall well supported.

Experimental design

The experimental setup is exhaustive and well designed in order to evaluate and validate the DePCR methods.

There is some inconsistencies in the methods text, and throughout the manuscript regarding the first step PCR: at some points 2 cycles, at other points 4 cycles are mentioned. As it must be 4 cycles for the setup to work, please correct the 2-cycle mentions to reflect the true experimental setup.

The evaluation of ´low´ DNA input at 1.25 ng per reaction is still quite a high input for many low biomass samples, to be honest. While I do not necessary think additional experiments are needed, the fact that low input DNA might be a limitation (even if many very low input samples are multiplexed before cleanup) should be discussed better. Also, the fact that a rather variable number of reads is retrieved, not directly correlating with DNA input, should be discussed in more detail.

Validity of the findings

The approach is very interesting, and a great improvement of the method previously published by this research group.
All presented conclusions are statistically well supported.

Additional comments

Considering the annealing temp gradient analysis, I think it would be interesting to present, at least as supplementary information, some compositional data (taxonomic classification of the obtained sequences/ASVs/OTUs) - to enable readers to have a glance at the biological consequences of the highly diverging annealing temperature effects between ´normal´PCR and DePCR PUP.

---

## Round 0.2 · Minor Revisions

line 39 Throughout, use an hyphen in PCR-amplified, or I would prefer we amplified XYX by PCR
Line 47, Can you add "distorting the original proportions in the original mixtures" before "extensive chimera formation"
Line 85, add « extracted as previously described » and the reference
Line 157, can you explain here whether the reads were demultiplexed before analysis and whether the primers were removed at this stage
line 165 there is a "by" missing before "this annotation"
line 168 "produce" instead of perform
line 325. Please add axis labels in figure S1C. Just by curiosity did you compare alpha diversity indices?
line 46, use "a single primer variant"
Line 466. Since you cite our paper, maybe I should make a comment here (even though the reviewers did not). When we talked about competition between primer and templates we also mean by the amplicons as templates, and the reason where at higher cycle numbers you would bias against "abundant" sequences. This is the reason I asked for the Shannon or even better Simpson indices differences between different starting concentrations, since you might see a higher evenness with more starting DNA (although the range of concentrations is not all that high).
line 524. You could suggest here that different second step reactions followed by normalization and pooling could be an alternative for those cases

·

Basic reporting

All fine

Experimental design

The revised methods are now easier to follow, in particular trough he improved Figure 2 and the corresponding legend.

Validity of the findings

All fine

Additional comments

Than you for your responses and revisions in response to my comments. I have no further comments to add.

---

## Round 0.3 · accepted · Accept

Thanks for answering all my queries, even just my "curiosity" question
Best regards